# Systemic Anti-Cancer Therapy and Metastatic Cancer Are Independent Mortality Risk Factors during Two UK Waves of the COVID-19 Pandemic at University College London Hospital

**DOI:** 10.3390/cancers13236085

**Published:** 2021-12-02

**Authors:** Yien Ning Sophia Wong, Christopher C. T. Sng, Diego Ottaviani, Grisma Patel, Amani Chowdhury, Irina Earnshaw, Alasdair Sinclair, Eve Merry, Anjui Wu, Myria Galazi, Sarah Benafif, Gehan Soosaipillai, Neha Chopra, Rebecca Roylance, Heather Shaw, Alvin J. X. Lee

**Affiliations:** 1Cancer Division, University College London Hospitals NHS Foundation Trust, London NW1 2BU, UK; sophia.wong@nhs.net (Y.N.S.W.); christopher.sng@nhs.net (C.C.T.S.); d.ottaviani@nhs.net (D.O.); grisma.patel1@nhs.net (G.P.); amani.chowdhury@nhs.net (A.C.); irina.earnshaw@nhs.net (I.E.); alasdair.sinclair@nhs.net (A.S.); eve.merry@nhs.net (E.M.); anjui.wu@nhs.net (A.W.); myria.galazi@nhs.net (M.G.); sarah.benafif@nhs.net (S.B.); g.soosaipillai@nhs.net (G.S.); neha.chopra@nhs.net (N.C.); r.roylance@nhs.net (R.R.); h.shaw2@nhs.net (H.S.); 2UCL Cancer Institute, University College London, London WC1E 6BT, UK; 3NIHR University College London Hospitals Biomedical Research Centre, London W1T 7DN, UK

**Keywords:** COVID-19, SARS-CoV-2 infection, solid cancer, risk factors, systemic anti-cancer therapy, co-morbidity, second wave, alpha variant, B.1.1.7

## Abstract

**Simple Summary:**

Cancer patients may have increased risk from COVID-19 due to impaired fitness and immunosuppression secondary to underlying cancer and the effects of anti-cancer treatments. We previously demonstrated that solid cancer and anti-cancer treatments may be associated with increased death following COVID-19 in an analysis of patients treated in our London hospital during the first wave of the COVID-19 pandemic (March to May 2020). The United Kingdom experienced a second peak of COVID-19 hospitalizations during December 2020 to February 2021. We aimed to compare the outcomes between patients with solid cancer presenting to our hospital during the first and second peaks of the COVID-19 pandemic and to determine if cancer and anti-cancer treatments were still risk factors for death. We found lower overall deaths in our hospital during the second peak. Metastatic cancer and anti-cancer treatments were risk factors for worse outcomes following COVID-19 in patients with cancer.

**Abstract:**

An increased mortality risk was observed in patients with cancer during the first wave of COVID-19. Here, we describe determinants of mortality in patients with solid cancer comparing the first and second waves of COVID-19. A retrospective analysis encompassing two waves of COVID-19 (March–May 2020; December 2020–February 2021) was performed. 207 patients with cancer were matched to 452 patients without cancer. Patient demographics and oncological variables such as cancer subtype, staging and anti-cancer treatment were evaluated for association with COVID-19 mortality. Overall mortality was lower in wave two compared to wave one, HR 0.41 (95% CI: 0.30–0.56). In patients with cancer, mortality was 43.6% in wave one and 15.9% in wave two. In hospitalized patients, after adjusting for age, ethnicity and co-morbidities, a history of cancer was associated with increased mortality in wave one but not wave two. In summary, the second UK wave of COVID-19 is associated with lower mortality in hospitalized patients. A history of solid cancer was not associated with increased mortality despite the dominance of the more transmissible B.1.1.7 SARS-CoV-2 variant. In both waves, metastatic disease and systemic anti-cancer treatment appeared to be independent risk factors for death within the combined cancer cohort.

## 1. Introduction

The Coronavirus disease 2019 (COVID-19) caused by the SARS-CoV-2 virus first peaked during the early months of 2020. Data from the initial wave across Europe, North America and Asia suggested that a history of cancer was associated with poorer outcomes with reported mortality rates ranging between 13 and 45% [1]. Cancer itself is associated with other SARS-CoV-2-related risk factors such as increasing age and certain co-morbidities. However, we [2] and others [3,4,5,6] have demonstrated that a history of solid malignancy was a risk factor for mortality when comparing patients with a history of cancer to matched patients without cancer during the first wave of COVID-19. Early concerns were also raised that the immunosuppressive effects of systemic anti-cancer treatment (SACT) such as cytotoxic chemotherapy could result in poorer outcomes in COVID-19 [7]. Subsequently, we demonstrated that SACT appeared to be an independent predictor of mortality following COVID-19 [2]. A meta-analysis of studies up to September 2020 showed that cytotoxic chemotherapy, but not targeted therapy or immunotherapy, was associated with increased mortality from COVID-19 [8].

In the United Kingdom (UK), a second peak of COVID-19 in the general population was observed from December 2020 to the early months of 2021. This surge of cases was linked to the discovery of the new, highly transmissible B.1.1.7 variant of SARS-CoV-2 in Southeast England [9]. This variant rapidly became the dominant variant of SARS-CoV-2 in the UK and in at least 114 countries [10]. While the B.1.1.7 variant appeared more transmissible, estimates suggested that it was not associated with increased mortality [10]. In addition to this UK variant, changes to clinical practice and public health measures likely had an impact on the spread of infection and outcomes of patients with COVID-19 in the second peak. Patients deemed clinically extremely vulnerable, due to their co-morbidities or ongoing treatment were recognized in a “Shielded Patient List” and advised to maintain strict social distancing measures by health authorities [11]. Due to the findings described above, these measures also encompassed patients with cancer undergoing active systemic anti-cancer treatment including chemotherapy. Meanwhile in community and healthcare settings, increased testing for COVID-19 has identified more patients with asymptomatic or mild illness [12,13]. Patients admitted with COVID-19 received European Medicines Agency (EMA) and Food and Drug Administration (FDA) approved treatments such as remdesivir [14], dexamethasone [15] and tocilizumab [16] following positive trials. Moreover, the UK national vaccination program commenced in December 2020 with those at risk prioritized for vaccination, including those of advanced age and having active cancer on SACT.

We set out to investigate the outcomes of patients with solid cancer compared to patients without cancer during the second peak of COVID-19 at University College London Hospitals. This is a tertiary hospital in central London incorporating a large cancer practice and providing acute medical care to both the local demographically diverse population and wider geography. We compared data from the second peak of COVID-19 (December 2020 to February 2021) to data from the first peak of COVID-19 (March 2020 to May 2020). Oncological variables such as patient demographics, cancer subtype, staging and anti-cancer treatment were also evaluated for association with COVID-19 mortality.

## 2. Materials and Methods

### 2.1. Ethics

In view of the retrospective nature of this study, approval from a Research Ethics Committee (REC) within the UK Health Departments Research Ethics Service and Health Research Authority (HRA) was not necessary. The criteria for REC review exemption were as follows: “Research involving previously collected, non-identifiable information”.

### 2.2. Data Collection

Anonymized retrospective data were collected from electronic health records of the University College London Hospitals NHS Foundation Trust across two time periods or waves coinciding with surges of COVID-19 infection in the UK. Patients aged 18 years or over presenting to our hospitals (via the emergency department or otherwise admitted as an inpatient) with a laboratory confirmed SARS-CoV-2 infection on RT-PCR from a throat/nose swab within each defined time period were included for initial screening. Patients with current or previously confirmed solid cancer were included. Age- and sex-matched patients within each respective wave, with no current or previous history of cancer, were selected randomly for comparison. Patients with concurrent hematological malignancies were excluded from the cancer and non-cancer cohort. Wave one included patients presenting between 1 March 2020 and 31 May 2020, and follow-up data were collected until 12 June 2020. Wave two included patients presenting between 1 December 2020 and 8 February 2021, and follow-up data were collected until 18 March 2021.

The primary outcome was all-cause mortality. Additional relevant data were collected for the cancer cohort. This included blood results at the time of diagnosis of SARS-CoV-2 infection, associated symptoms and cancer-specific data.

### 2.3. Definitions

Active cancer was defined as a solid tumor cancer diagnosis or anti-cancer treatment within the last 12 months or radiological or biochemical evidence of active or recurrent cancer. Active anti-cancer therapy was defined as any anti-cancer therapy within 60 days of the COVID-19 diagnosis, including SACT, radiotherapy and surgery. SACT included all systemic anti-cancer treatment modalities such as cytotoxic chemotherapy, endocrine therapy, immunotherapy and targeted anti-cancer therapy. A composite co-morbidity score was calculated for patients by assigning a weight of 1 to each co-morbidity including a history of cancer recorded in this study and summing the total for each patient.

### 2.4. Statistical Analysis

Descriptive statistics was utilized on baseline demographic data between patient cohorts. Continuous data are presented as median and interquartile ranges (IQR) unless otherwise specified. Categorical data are shown as frequency and percentage. Mann–Whitney U tests were used to compare continuous data, and Fisher’s exact test or Chi-Square test were used for categorical variables.

As described in our previous publication [2], Cox proportional-hazards regression model was used to evaluate overall survival from the date of COVID-19 diagnosis. Univariate Cox regression was used to obtain crude hazard ratios for each pre-morbid risk factor within each cohort. Three multivariate models were developed in planned analyses for this study. The first two included all patients in each respective wave. Risk factors that were statistically significant (uncorrected *p*-value < 0.05) at univariate analysis in each respective wave were adjusted for in a multivariate model to assess the independent impact of cancer history in each wave. In order to study the effect of recent SACT, subgroup analysis was performed on all cancer patients from wave one and two. Similarly, significant variables at univariate analysis of this cancer cohort were adjusted for in a single multivariate model to elucidate the effect of SACT on overall survival. All survival analyses were conducted using the survival package in R 3.6.1.

## 3. Results

### 3.1. Demographics of Patients with Cancer and Patients without Cancer in Both Waves of COVID-19

Among patients presenting to University College Hospitals (via an emergency department assessment or other admission routes to hospital) with a confirmed SARS-CoV-2 infection, there were 1135 in wave two (1 December 2020 and 8 February 2021, 70-day period) and 626 in wave one (1 March 2020 to 31 May 2020, 92-day period). These data collection periods corresponded to consecutive waves in COVID-19 infection and hospitalization in the UK (Figure 1). 

Across both waves, a total of 207 patients with solid cancer were age- and sex-matched to 452 patients without cancer that were selected randomly for comparison. There were 94 (10.0%) patients with solid cancer in wave one and 113 (15.0%) patients with solid cancer in wave two in all confirmed SARS-CoV-2 patients presenting to the hospital. The combined median follow-up time from the date of COVID-19 diagnosis was longer in wave two versus wave one (31 days, IQR 11–51, versus 18 days, IQR 7.8–44). Table 1 summarizes the demographic features of the solid cancer and non-cancer cohorts in both waves.

Overall, hospitalization and mortality rates were significantly lower in wave two versus wave one (*p* < 0.001) at follow-up, despite almost double the number of a laboratory confirmed SARS-CoV-2 infection in wave two. A breakdown by wave and cancer history is shown in Table 1. There were more South Asian, fewer White and fewer male patients in wave two compared to wave one. Patients in wave two appeared to have fewer co-morbidities when comparing a composite co-morbidity score (Table 1). Among these, there were fewer patients with dementia, peripheral vascular disease and cerebrovascular disease in wave two. Patients with cancer had a similar level of co-morbidity as shown by their composite co-morbidity score (2.79 versus 2.81).

Most patients in wave two received approved COVID-19 specific treatments including dexamethasone (50.4%), remdesivir (18.0%), tocilizumab (4.1%). Among those with a history of cancer, 32.7% of patients received dexamethasone, 8% received remdesivir and one patient received tocilizumab. As expected, a significantly lower number of patients received any of these drugs in wave one (*p* < 0.0001). A similar proportion of patients with solid cancer were admitted to an intensive therapy unit (ITU) or high dependency unit (HDU) in wave two compared to wave one (21.2% versus 23.4%). The incidence of invasive and non-invasive ventilation was similar among patients with cancer in wave two (8.8% and 15.0%, respectively) and wave one (5.3% and 13.8%, respectively). Among patients with active solid cancer, the rates of ITU/HDU admission in wave two and wave one were 22.4% and 20.0%, respectively.

### 3.2. Risk Factors of SARS-CoV-2 Mortality in Both Waves

Overall survival following SARS-CoV-2 infection significantly improved in wave two compared to wave one in all presenting patients (*p* < 0.0001) and hospitalized patients (*p* < 0.0001) (Figure 2). The hazard ratio for death in wave two versus wave one in all patients was 0.41 (95% CI: 0.30–0.56). A meaningful comparison of outcomes in wave one and two was complicated by wider testing and lower acuity at presentation in wave two. In order to mitigate this, we focused our analysis on patients who were already inpatients or hospitalized with COVID-19.

Univariate analysis of the hospitalized cancer and non-cancer cohorts demonstrated that increasing age, hypertension and cardiovascular disease were common factors associated with mortality in both wave one and wave two (Figure 3). In wave two, diabetes, congestive cardiac failure and male sex were also significantly associated with mortality. Among all hospitalized patients, there was a trend towards improved survival for those receiving COVID-19 specific treatment including dexamethasone, remdesivir and tocilizumab (HR 0.75, 95% CI: 0.55–1, *p* = 0.086).

A multivariate survival analysis was performed to assess the independent contribution of cancer to mortality risk following SARS-CoV-2 infection in hospitalized patients, adjusting for age and/or sex and comorbidities that were significant in the univariate analysis in each wave (Figure 4). A history of solid cancer was an independent risk factor for mortality amongst patients hospitalized with COVID-19 in wave one (HR 1.62, 95% CI: 1.07–2.5, *p* = 0.02) but not in wave two (HR 1.00, 95% CI: 0.57–1.8, *p* = 0.97). Other risk factors in wave two, which remained significant at multivariate analysis, included male sex (HR 2.4, 95% CI: 1.32–4.4, *p* = 0.004) and age (HR 1.4 for every 10 years, 95% CI: 1.07–1.7, *p* = 0.01).

### 3.3. Determinants of Mortality among Patients with Cancer from Both Waves

Out of the 207 patients with solid cancer across both waves (both hospitalized and non-hospitalized), the most common cancer type was gastrointestinal (n = 52), followed by genitourinary (n = 51), thoracic (n = 28), breast (n = 22) and gynecological cancer (n = 20). The distribution of cancer type and proportion of patients with active cancer were similar in both waves (61.9% and 61.7%) (Appendix A). Likewise, there was no statistically significant difference in the proportion of patients with metastatic disease in wave two versus wave one (24.8% versus 20.2%, respectively, *p* = 0.42).

Across both waves, 58 (30.1%) patients were actively receiving SACT at the time of COVID-19 diagnosis. This included chemotherapy (n = 40), endocrine therapy (n = 15), targeted anti-cancer therapy (n = 8) and immunotherapy (n = 5). 10 patients received a combination of different SACT modalities, with the majority receiving chemotherapy in combination with immunotherapy (n = 4), targeted therapy (n = 3) or endocrine therapy (n = 2). 35 patients were being treated with palliative intent. There were too few primary outcomes reached in the cancer cohort of wave two for any meaningful analysis in this study. Therefore, we combined data from both waves to assess the impact of SACT on mortality. In order to evaluate whether SACT is an independent risk factor for mortality following COVID-19, we performed multivariate survival analysis, adjusting for factors that were significant at univariate analysis in the combined cancer cohort (male sex, hypertension, cerebrovascular disease, age and presentation in wave two) (Appendix A). 

Receipt of SACT remained an independent risk factor for COVID-19 mortality in the combined cancer cohort (HR 2.01, 95% CI: 1.10–3.66, *p* = 0.02) (Figure 5), as well as in the hospitalized cancer cohort (HR 2.14 95% CI:1.19–3.84, *p* = 0.01). Multivariate survival analysis, limited by small numbers, demonstrated a trend towards significance for cytotoxic chemotherapy as an independent predictor of mortality with a HR 1.93 (95% CI: 0.93–4.00, *p* = 0.08), while the other individual treatment modalities were not associated with significant differences in COVID-19 mortality (Table 2). We also found that patients with metastatic disease were found to have a higher mortality following COVID-19 infection (HR 2.1, 95% CI: 1.02–4.34, *p* = 0.04) (Table 2). Only 3 out of 47 (6.4%) patients with metastatic disease were admitted to ITU/HDU, compared to 40 out of 160 (25.0%) patients with non-metastatic disease (*p* = 0.004). Seventeen patients with metastatic disease died following COVID-19, and only one of these patients were admitted to ITU/HDU.

## 4. Discussion

To our knowledge, this is one of the first reports on outcomes in patients with solid cancer encompassing the first and second waves of the COVID-19 pandemic in a UK-based cohort. A number of other studies compared the first and second waves but did not focus on hospitalized patients with cancer and were not UK-based [17,18]. We compared outcomes following SARS-CoV-2 infection in patients with solid cancer compared to a cohort of contemporaneous matched patients without cancer. Strikingly we found that the risk of death following SARS-CoV-2 infection among patients presenting to the hospital was significantly lower during the second peak compared to the first peak. Hospitalization rates were also significantly lower in the second wave. The absolute increase in SARS-CoV-2 infected patients in wave two likely reflected not only increased infection rates in the population but also increased testing and lower thresholds for presentation to secondary care. In order to mitigate this, we focused our analysis on hospitalized patients. Patients with solid cancer did not have an increased risk of mortality in wave two, unlike in wave one. Our defined time-period of wave two coincided with a rapid rise in the B.1.1.7 variant of SARS-CoV-2, accounting for over 95% of positive PCR results in England by the end of our inclusion period [19], suggesting that the new variant did not pose excess risk to those with cancer from this single institution cohort. During April 2021, the B.1.1.7. variant was the dominant variant in England and had been reported in 137 countries [19].

The overall improved survival in wave two could be due to a combination of factors including but not limited to the following: COVID-19 vaccination, diversion of hospital resources to manage COVID-19, introduction of COVID-19-specific therapies, widespread testing allowing earlier diagnosis and treatment, shielding and COVID-19-protected patient pathways. The largest contributor to better outcomes in wave two is likely to be COVID-19 vaccination. Recent studies have demonstrated that patients with COVID-19 develop adequate levels of antibodies and incidence of COVID-19 infection and serious disease was reduced in patients with cancer who have received at least one dose of a COVID-19 vaccine [20,21]. In the UK, patients with cancer were prioritized to receive the vaccine from December 2020 [22]. This is likely to have contributed significantly to improved outcomes observed in patients with cancer, and early prioritization of patients with cancer for vaccinations was a strategy replicated in other countries [23]. Additionally, during the first wave in the UK, patients with cancer appeared to be less likely received COVID-19-specific therapy such as dexamethasone, which may have contributed to poorer outcomes for patients with cancer [24]. Finally, in the absence of shielding, more vulnerable patients could have acquired COVID-19 in wave one, resulting in worse outcomes in that wave. Following the first COVID-19 wave, there could also be a level of population and individual immunity to COVID-19 during the second wave which may have been protective in wave two [25].

Patients with cancer appear to have the worse outcomes following COVID-19 [1,2,26], with higher mortality observed in patients admitted to hospital or to intensive care units (ICU). Cancer disease status appears to be associated with outcomes, with those receiving curative treatments having better outcomes [27]. It is less clear if SACT and cytotoxic chemotherapy independently contribute to poorer outcomes following COVID-19. We have found a consistent association between recent SACT and death in our analysis of patients with cancer in wave one and two. Recent data from North America [28] and a meta-analysis [8] also support the association of cytotoxic chemotherapy but not other modalities of SACT with poorer outcomes. However, two large studies from a wave one cohort of patients with cancer and COVID-19 in a UK or predominantly UK patient population [29,30,31,32] did not observe an association between recent chemotherapy and increased risk of death. This could be explained by variance in patient population including ethnicity and oncological and general clinical practices between centers and less complete follow-up data compared to our dataset. Differences in the temporal definition of recent chemotherapy may also contribute variation in the effect of treatment on COVID-19 outcomes. One study defined active treatment as falling within the preceding 12 months of a COVID-19 diagnosis and did not find a significant effect on mortality [31]. Meanwhile another study defining active treatment as falling within 3 months found that cytotoxic chemotherapy was associated with poor COVID-19 outcomes [28]. Patient selection may also explain the lack of association between SACT and mortality in these early studies as patients who were younger, fitter and on earlier lines of treatment were prioritized for treatment with SACT. Moreover, UK national guidance was issued early in the first wave of the UK COVID-19 pandemic. This guidance included advice on prioritization of patient groups who were fitter and more likely to benefit from treatment and modulation of SACT treatments to those that were less immunomodulatory or that required less contact time in healthcare settings [33]. These changes in clinical practice may explain the failure to detect an association between chemotherapy and mortality from studies using data from the first wave.

We did not detect a difference in mortality in patients who had received immunotherapy. This is likely as we only had five patients on immunotherapy who developed COVID-19 across our patient cohorts and of which four received immunotherapy in combination with chemotherapy. However, a meta-analysis comprising 3581 cancer patients with COVID-19 did not detect an increased risk of mortality for patients receiving immunotherapy [34]. They, however, did report increased severe events following COVID-19 in patients who received immunotherapy within 90 days of developing COVID-19. Immunotherapy can also result in pneumonitis, an immune-related adverse event (irAE). Given the diminished respiratory reserve of lung cancer patients due to cancer and the widespread use of immunotherapy in this disease, this may place these patients at increased risk of severe outcomes or mortality following COVID-19. However, the TERAVOLT [35] study, which analyzed patients with lung cancer, did not observe an increase in mortality following immunotherapy.

Therefore, currently, there does not appear to be strong evidence that immunotherapy causes harm in cancer patients who develop COVID-19. Conversely, there have been recent hypotheses putting forth the viewpoint that immunotherapy may potentially be beneficial to patients following COVID-19 [36,37]. The authors argue that cytokine release syndrome, which can occur following COVID-19, is a hyper-inflammatory first phase following infection with results in T-cell hyperactivation and exhaustion. Subsequently, PD-1/PD-L1 or CTLA-4 blockade may invigorate T-cell viral control responses. An ongoing clinical trial (NCT04333914) aims to explore this hypothesis further by investigating differential efficacy to eradicate SARS-CoV-2 infection in COVID-19 patients treated either with anti-PD-1 antibody nivolumab in association with standard care protocol or with standard care alone. 

## 5. Conclusions

The overall mortality regardless of a history of cancer for hospitalized patients in our institution improved in the second UK wave of COVID-19. These findings affirm the steps in improving COVID-19 vaccine availability for vulnerable patients and in increasing resources for COVID-19 detection, shielding, prevention and new treatments. A history of solid cancer reassuringly did not appear to contribute to increased mortality despite the dominance of the more transmissible B.1.1.7 SARS-CoV-2 variant during the second wave, substantiating focused efforts to protect those with cancer. When combining patients with solid cancer from both waves, SACT and metastatic disease appeared to be independent predictors of mortality following COVID-19. This suggests that the immunomodulatory effects of treatment and trajectory of disease are key determinants of outcome following COVID-19 for those with cancer. These results emphasize the need for ongoing protection of patients with advanced cancer and those on SACT, including their prioritization for COVID-19 vaccination and boosters worldwide.

## Figures and Tables

**Figure 1 cancers-13-06085-f001:**
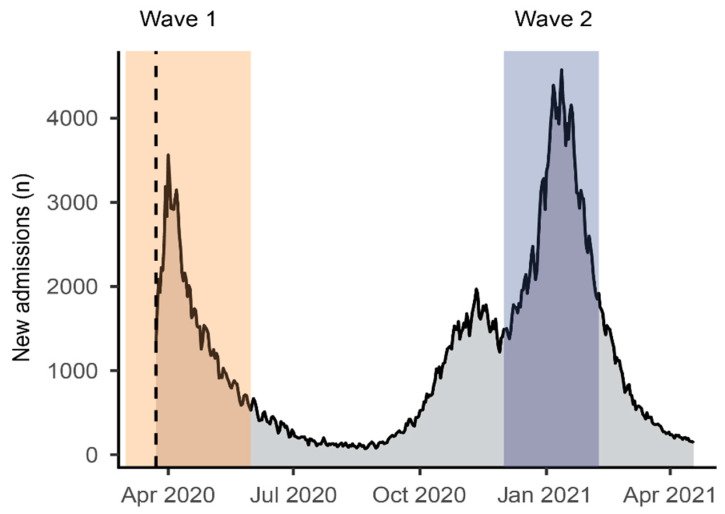
Daily new admissions in the UK of patients with COVID-19 with corresponding study periods. Shaded rectangles indicate this study’s inclusion time periods. The dotted line represents the start of publicly available data. Data from https://coronavirus.data.gov.uk/details/healthcare, accessed on 22 April 2021.

**Figure 2 cancers-13-06085-f002:**
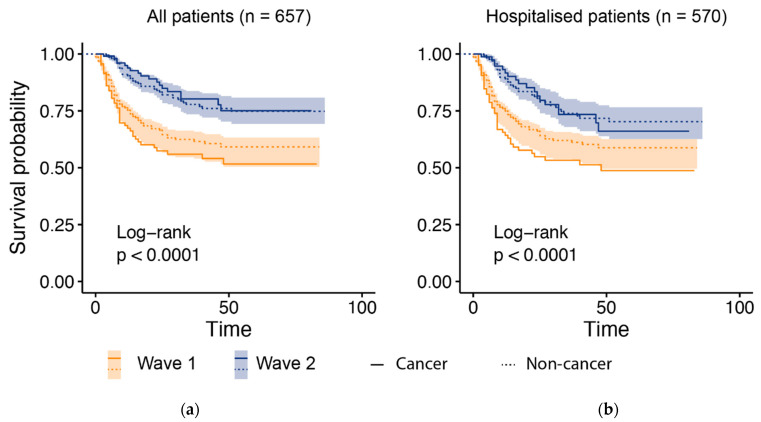
Overall survival in wave two was significantly improved compared to wave one in patients with or without cancer. The shaded regions show 95% CI for the survival probability of all patients in each respective wave for all presenting patients (**a**) and hospitalized patients (**b**) with or without cancer, and a Log-rank test was used to compare survival curves for all patients in each respective wave. Overall survival in wave two was significantly improved compared to wave one in patients with or without cancer.

**Figure 3 cancers-13-06085-f003:**
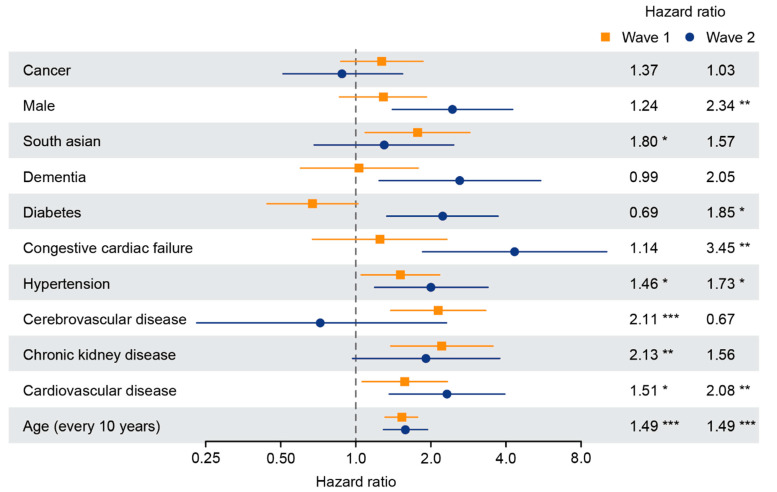
Univariate risk factors for COVID-19 mortality in wave one and wave two in hospitalized patients with or without cancer. Forest plot showing the hazard ratios from univariate analyses of risk factors associated with mortality in COVID-19. Horizontal bars indicate 95% confidence interval. *, *p* < 0.05; **, *p* < 0.01; ***, *p* < 0.001.

**Figure 4 cancers-13-06085-f004:**
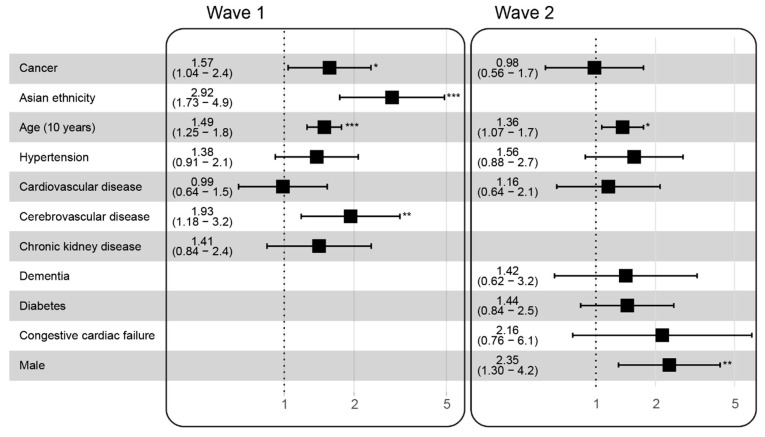
Comparison of multivariate models for hospitalized patients in wave one versus wave two. Age was the single most consistent risk factor for mortality to COVID-19 in both waves in multivariate modelling. Adjusted hazard ratios with 95% confidence intervals are shown for the variables included in each multivariate model. Variable selection for each model was based on statistical significance (*p* < 0.05) at univariate analysis (see Figure 3). *, *p* < 0.05; **, *p* < 0.01; ***, *p* < 0.001.

**Figure 5 cancers-13-06085-f005:**
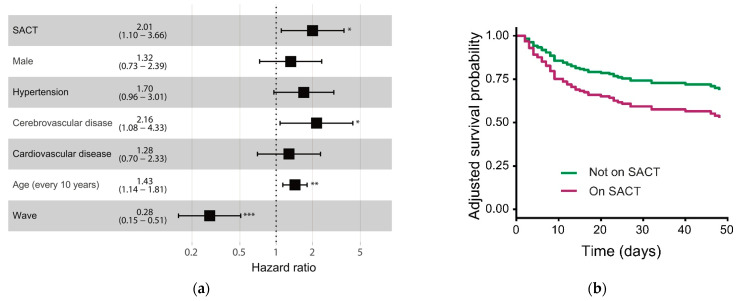
Systemic anti-cancer therapy is an independent predictor of mortality in cancer patients following COVID-19 in both waves. (**a**) Multivariate survival analysis of risk factors in patients with cancer and COVID-19. Adjusted hazard ratios with 95% confidence intervals are shown. (**b**) Adjusted survival curves of patients on systemic anti-cancer therapy with COVID-19. *, *p* < 0.05; **, *p* < 0.01; ***, *p* < 0.001.

**Table 1 cancers-13-06085-t001:** Demographic data of patients with COVID-19 stratified by wave and cancer history. Data are shown as n (%) or median (IQR) unless otherwise specified. *p*-value significance level is indicated for the combined cancer and non-cancer cohort in each wave. Legend: BMI, body mass index; †, data shown as mean (standard deviation); * *p* < 0.05, ** *p* < 0.01, *** *p* < 0.001, **** *p* < 0.0001.

	Wave 1 Control (n = 226)	Wave 1 Cancer (n = 94)	Wave 2 Control (n = 226)	Wave 2 Cancer (n = 113)	*p*-Value
Male	152 (67.3%)	62 (66.0%)	122 (54.0%)	61 (54.0%)	***
Median age(years)	70.50 (60.00–80.00)	71.00 (62.00–80.00)	71.00 (58.00–79.00)	71.00 (58.00–79.00)	
BMI	26.59 (23.45–30.47)	25.05 (21.72–30.48)	26.71 (23.34–32.12)	26.20 (22.37–30.48)	
South Asian	28 (12.4%)	8 (8.5%)	42 (18.6%)	18 (15.9%)	**
Black	37 (16.4%)	6 (6.4%)	29 (12.8%)	10 (8.8%)	
Other	19 (8.4%)	9 (9.6%)	17 (7.5%)	13 (11.5%)	
White	115 (50.9%)	64 (68.1%)	78 (34.5%)	54 (47.8%)	**
Ex or active smoker	71 (31.4%)	49 (52.1%)	64 (28.3%)	47 (41.6%)	
Cardiovascular disease	59 (26.1%)	18 (19.1%)	57 (25.2%)	19 (16.8%)	
Dementia	35 (15.5%)	7 (7.4%)	17 (7.5%)	7 (6.2%)	**
Diabetes	73 (32.3%)	24 (25.5%)	77 (34.1%)	25 (22.1%)	
Congestive cardiac failure	16 (7.1%)	9 (9.6%)	11 (4.9%)	4 (3.5%)	
Liver disease	4 (1.8%)	3 (3.2%)	5 (2.2%)	0 (0.0%)	
Hypertension	123 (54.4%)	37 (39.4%)	106 (46.9%)	48 (42.5%)	
Peripheral vascular disease	13 (5.8%)	2 (2.1%)	1 (0.4%)	4 (3.5%)	*
Cerebrovascular disease	37 (16.4%)	12 (12.8%)	12 (5.3%)	14 (12.4%)	**
Chronic lung disease	47 (20.8%)	14 (14.9%)	51 (22.6%)	16 (14.2%)	
Chronic kidney disease	26 (11.5%)	12 (12.8%)	23 (10.2%)	13 (11.5%)	
Ongoing corticosteroids	13 (5.8%)	4 (4.3%)	2 (0.9%)	3 (2.7%)	**
Composite co-morbidity score †	2.57 (1.71)	2.79 (1.44)	2.04 (1.33)	2.81 (1.42)	*
COVID-19-specific therapy	13 (5.8%)	1(1.1%)	134 (59.3%)	37 (32.7%)	****
Hospitalization	223 (98.7%)	86 (91.5%)	179 (79.2%)	82 (72.6%)	***
Intensive care admission	76 (33.6%)	22 (23.4%)	68 (30.1%)	24 (21.2%)	
Death	77 (34.1%)	41 (43.6%)	43 (19.0%)	18 (15.9%)	***

**Table 2 cancers-13-06085-t002:** Systemic anti-cancer therapy and metastatic disease are independent risk factors in cancer patients to COVID-19 mortality from both waves. Multivariate survival analysis comparing malignancy status and recent types of anti-cancer treatment in patients with cancer and COVID-19. Table shows the respective hazard ratio (HR) with 95% confidence interval (CI) and *p*-value.

	HR (95% CI)	*p*-Value
**Malignancy status**		
Metastatic	**2.1 (1.02–4.34)**	**0.04**
**Active anti-cancer treatment**	1.75 (0.97**–**3.18)	0.06
SACT	**2.01 (1.10–3.66)**	**0.02**
Cytotoxic chemotherapy	1.93 (0.93**–**4.00)	0.08
Endocrine therapy	1.66 (0.69**–**3.96)	0.25
Targeted therapy	0.84 (0.11**–**6.28)	0.86
Immunotherapy	1.73 (0.4**–**7.41)	0.46
Radiotherapy	2.04 (0.62**–**6.74)	0.24
Surgery	0.67 (0.09**–**4.98)	0.69

## Data Availability

The datasets presented in this article are not readily available as all COVID-19 related data access require permission from the University College London Hospitals Data Access Committee. Requests to access the datasets should be directed to uclh.randd@nhs.net.

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
