# Peer review of "Systemic Anti-Cancer Therapy and Metastatic Cancer Are Independent Mortality Risk Factors during Two UK Waves of the COVID-19 Pandemic at University College London Hospital"

_cancers, 2021, doi:10.3390/cancers13236085_

Round 1

Reviewer 1 Report

In this study, Wong, et al. conducted retrospective analysis encompassing two waves of COVID-19 in the UK. Although the study was mostly descriptive and performed at single center, it demonstrated the association between cancer (-related parameters) and COVID-19, which could be helpful for health professionals to understand COVID-19 and cancers.

Author Response

Thank you for your feedback on our manuscript. Formatting and spelling has been checked in this version and we do not detect further problems in this respect.

Reviewer 2 Report

Wong and colleagues presented a retrospective observational study aimed at assessing cancer patients’ mortality during the first and second wave of COVID-19 infection in the UK. For this purpose, the authors evaluated cancer patients’ mortality in wave 1 and wave 2 compared to the mortality rates observed in a control cohort of non-cancer patients. In addition, the authors took into account patients’ comorbidities as well as anticancer therapies. Overall, the manuscript is interesting and new, however, there are some issues that the authors have to address before publication. Below are reported some minor/major comments that will improve the quality of the manuscript:
1) How do the authors explain the lower overall deaths observed during the second peak? May it be due to a better awareness of COVID-19 infection and more appropriate treatments? Please clarify and discuss better this point in the Discussion section;
2) Please remove the following sentence in the Results section: “This section may be divided by subheadings. It should provide a concise and precise description of the experimental results, their interpretation, as well as the experimental conclusions that can be drawn.”;
3) The following title is repeated twice in the manuscript. Please fix this issue: “3.1. Demographics of patients with cancer and patients without cancer in both waves of COVID-19”;
4) For a better description of COVID-19 incidence in cancer patients during the first and second wave of infection it could be useful to provide data of all patients admitted in both waves. In particular, among all covid patients, how many had a tumor (express the data as a percentage of the total covid-19 hospitalized patients)?;
5) The authors demonstrated that cancer patients in wave 1 have higher mortality compared to patients in wave 2 (mortality rate passes from 44% in wave 1 to 16% in wave 2). Considering this statistically significant difference between patients’ mortality in wave 1 and wave 2 it is not clear why in chapter 3.3 the authors do not stratify cancer patients in wave 1 and wave 2 to establish reliable HR calculated according to different anti-cancer treatments. Indeed, the data reported in table 2 are related to both cohorts of cancer patients (wave 1 and wave 2) despite these two cohorts have completely different mortality rates. This represents a major analytical bias. I understand that separating the two cohorts of patients further reduces the number of cases, however, merging patients from wave 1 and wave 2 is not statistically correct due to the different rates of mortality. Please address these issues;
6) Please revise table 2. Remove the columns “Title 1”, “Title 2” and “Title 3” if unnecessary;
7) Check the error “focussed” in the following sentence: “To mitigate this, we focussed our initial analysis on hospitalised patients.”;
8) In the second part of the Discussion section, the authors describe their data and literature data about the impact of anti-cancer therapies on patients’ survival. Based on my previous comment No. 5, the authors have to better argue this part of the Discussion section, as several studies have demonstrated that surgery, chemotherapy and radiotherapy represent risk factors for cancer patients while immunotherapy seems to be protective in some cases. For this purpose, please better describe these aspects and see recent literature reports. Please, see:
- PMID: 33491759
- PMID: 32473682
- PMID: 33294299
- PMID: 32785162
- PMID: 34406802
- PMID: 33758663

Author Response

Thank you for your comprehensive feedback. We have taken onboard your comments and we have addressed them as follows:

1) Thank you, we have added to our discussion regarding better outcomes in this patient population. We feel that our data and subsequent data suggests that better outcomes would have been predominantly driven by vaccination and in the UK cancer patients were prioritised to receive COCVID-19 vaccines from December 2020 onwards.

“The overall improved survival in wave 2 could be due to a combination of factors including, but not limited to; COVID-19 vaccination, diversion of hospital resources to manage COVID-19, introduction of COVID-19-specific therapies, widespread testing allowing earlier diagnosis and treatment, shielding and COVID-19-protected patient pathways. The largest contributor to better outcomes in wave 2 is likely to be COVID-19 vaccination. Recent results presented at ESMO Congress 2021 demonstrate that patients with COVID-19 develop adequate levels of antibodies and incidence of COVID-19 infection and serious disease is reduced in patients with cancer who have received at least one dose of a COVID-19 vaccine [20,21]. In the UK, patients with cancer were prioritized to receive the vaccine from December 2020 [22]. This is likely to have contributed the most to the improved outcomes seen in patients with cancer and was a strategy replicated in other countries [23]. Additionally, during the first wave in the UK, patients with cancer appeared to be less likely to have received COVID-19-specific therapy such as dexamethasone which may have contributed to poorer outcomes for patients with cancer [24]. Finally, more vulnerable patients could have acquired COVID-19 in wave 1, resulting in worse outcomes in that wave. Following the first COVID-19 wave, there could also be a level of population and individual immunity to COVID-19 during the second wave which may have been protective in wave 2.”

2) “This section may be divided by subheadings. It should provide a concise and precise description of the experimental results, their interpretation, as well as the experimental conclusions that can be drawn.”; These lines from 139 to 141 which originated from the template has been deleted

3) “3.1. Demographics of patients with cancer and patients without cancer in both waves of COVID-19”; - the repeated duplicate second subsection heading has been deleted.

4) For clarity we have now included percentages so it will be clearer to the reader what the percentage of patients with cancer amongst all patients with COVID-19 for both waves.

“There were 94 (10.0%) patients with cancer in wave 1 and 113 (15.0%) patients with cancer in wave 2 in all confirmed SARS-CoV-2 patients presenting to hospital.”

5) There were too few events in wave 2 to stratify by anti-cancer analysis. Therefore, we combined data from both waves for SACT analysis. To evaluate whether SACT as an independent risk factor for mortality following COVID-19, we performed multivariate survival analysis, adjusting for risk factors which were significant at univariate analysis in the combined cancer cohort (wave 2, age) (Supplementary Table 2). The multivariate analysis including the variable “wave” which adjusted for effects of wave 1 and wave 2.

“There were too few events in wave 2 for any meaningful analysis. Therefore, we combined data from both waves for SACT analysis.” – line 234 to 236

6) Table 2 extra columns which originated from merging the formatting into the template has been deleted

7) This spelling mistake has been corrected. As the template uses English (US), we have changed all spellings in our manuscript to English (US) and have not detected any further spelling mistakes in the manuscript.

8) We have added to our discussion addressing other studies looking at the impact of chemotherapy and immunotherapy to cancer patients who develop COVID-19, with further citations looking at more recent publications.

“Patients with cancer appear to have worse outcomes following COVID-19 [26] [1,2] with higher mortality seen in patients admitted to hospital or to intensive care units (ICU). Cancer disease status appears to be associated with outcomes, with those receiving curative treatments having better outcomes [27]”

”Recent data from North America[28] and a meta-analysis [8] also support the association of cytotoxic chemotherapy, but not other modalities of SACT, with poorer outcomes. However, two large studies from a wave 1 cohort of patients with cancer and COVID-19 in a UK or predominantly UK patient population [29-32] did not see an association between recent chemotherapy and increased risk of death.”

“We did not detect a difference in mortality in patients who had received immunotherapy. This is likely as we only had 5 patients on immunotherapy who developed COVID-19 across our patient cohorts, and of which 4 received immunotherapy in combination with chemotherapy. However, a meta-analysis comprising 3581 cancer patients with COVID-19 did not detect an increased risk of mortality for patients receiving immunotherapy [34]. They however did report increased severe events following COVID-19 in patients who received immunotherapy within 90 days of developing COVID-19. Immunotherapy can also result in pneumonitis, an immune-related adverse event (irAE). Given the diminished respiratory reserve of lung cancer patients due to their cancer and the widespread use of immunotherapy in this disease, this may put these patients at increased risk of severe outcomes or mortality following COVID-19. However, the TERAVOLT [35] study which analyzed patients with lung cancer, did not see an increase in mortality following immunotherapy.

Therefore, currently there does not appear to be strong evidence that immunotherapy causes harm in cancer patients who develop COVID-19. Conversely, there have been recent hypotheses putting forth the viewpoint that immunotherapy may be potentially beneficial to patients following COVID-19 [36,37]. The authors argue that cytokine release syndrome which can occur following COVID-19 is a hyper-inflammatory first phase following infection with results in T-cell hyperactivation and exhaustion. Subsequently, PD-1/PD-L1 or CTLA-4 blockade may invigorate T-cell viral control responses. An ongoing clinical trial (NCT04333914) aims to explore this hypothesis further by investigating differential efficacy to eradicate SARS-CoV-2 infection in COVID-19 patients treated either with anti-PD-1 antibody nivolumab in association with standard care protocol, or with standard care alone.”

Reviewer 3 Report

This work is describe systemic Anti-Cancer Therapy and Metastatic Cancer are independent mortality risk factors during two UK waves of the COVID-19 pandemic at University College London Hospital. It is very interesting topic. However, the novelty of this manuscript is seemed to be weakly.

  1. The manuscript was suggested to check carefully (ex; Table 2, line 139-141 et al).
  2. To calculate the mortality risk factors,  data collected may be biased as survival vary between cancer types.
  3. Vaccination (COV-19 related vaccines) was suggested to add in the Table 1. 

Author Response

Thank you for your constructive comments and giving us the opportunity to improve our manuscript. To our knowledge, this remains one of the first reports directly comparing outcomes of patients with cancer during the first and second peaks of the COVID-19 pandemic and certainly one of the first such manuscripts in the UK, with the second peak and beyond of COVID-19 characterised by emergence of SARS-CoV2 variants, and availability drug treatments and the rollout of vaccines, where in the UK, patients with cancer were a priority group. Therefore, we feel that it is important for us to share our findings with the wider research community. We have added the following to our discussion to address this “To our knowledge, this report is one of first to report on outcomes in patients with cancer encompassing the first and second waves of the COVID-19 pandemic in a UK-based cohort with other studies comparing first and second waves not focusing on hospitalized patients with cancer and no UK-based patient cohort [17,18].”

  1. Line 139 to 141 which originated from the template has been deleted.

Table 2 extra columns which originated from merging the formatting into the template has been deleted.

We have changed the language to US English to match the template and have not detected further spelling mistakes.

  1. Supplementary Table 1 provides numbers of patients for each cancer sub-type. And Supplementary Table 2 contains univariate analysis by cancer type, we did not detect a difference in survival between cancer subtypes but this could have been limited by patient number for each subtype.
  2. Vaccination status was not routinely recorded within the electronic record system at the time of analysis. Vaccinations were not available during wave 1 of the pandemic. In the UK, cancer patients were prioritised to receive vaccinations from December 2020. “In the UK, patients with cancer were prioritized to receive the vaccine from December 2020 [22]” – line 368.

Reviewer 4 Report

Wong and colleagues present data on a heterogeneous cohort of patients with solid tumors and COVID-19 during two waves in a UK-based university hospital. Risk factors for mortality were analysed showing systemic anti-cancer therapy and metastatic disease as associated with increased mortality in cancer patients. While these risk factors are well known, epidemiological results on differences between wave 1 and 2 with regard to cancer patients are of interest.

Major comments include:

  • It is unfortunate that patients with haematological malignancies were excluded from the cancer cohort as they seem to be a population at particular risk. Exclusion of these patients might very well have significantly influenced any impact of cancer on mortality. It should therefore stated clearly at all instances, most importantly in the abstract, that only solid tumor patients were addressed in the cancer cohort, and possible implications should be discussed.
  • analyses of all patients and of hospitalized patients only are not in all instances clearly distinguished. It should be made clear, most importantly inthe abstract, which population is addressed.
  • Effect of different COVID treatments should be assessed separately and more prominently included in statistical analyses.
  • Vaccination status and/or prior infection might have influenced mortality in wave 2. This data should be presented and incorporated into statistical analyses.
  • In solid cancer patients, time since diagnosis seem to exert a major influence on COVID-19 moratility (Williamson Nature 2020). This data should be presented and incorporated into statistical analyses.

Minor comments include:

  • 108: please state until what time after COVID diagnosis all cause mortality was considered as an endpoint
  • 108: please state what "additional relevant data" was collected
  • 114: 60day period uncommonly long to address active treatment, 30 day period might be more appropriate, please discuss
  • 139-141: probably forgotten to delete the instructions to authors, please delete.
  • 190: it is stated that patients in wave 2 were less symptomatic at presentation although no data is presented to support this. Symptoms and severity at presentation should be added.
  • 255: empty columns to be deleted.

Author Response

Thank you for your constructive comments, we hope you will find this revised manuscript acceptable.

  1. We only included patients with solid cancers in our analysis as we are reporting from the solid cancer division in our hospital. Where appropriate including in the abstract and elsewhere where it may cause ambiguity, we have used the term “solid cancers” instead of just cancer, so as to be clear we do not include haematological malignancies in our analysis (patients with haematological malignancies were excluded from our control group). This is also reflected in our inclusion and exclusion criteria in the methods section.
  2. We have made in clearer in the abstract, where we look at overall patients and where we focus on hospitalized patients. Results section 3.2 onwards focus on hospitalized patients. Figure legends and text have been modified to make this clearer. Results 3.3 is in all cancer patients. Again, text has been modified to improve clarity.
  3. As this study was not designed to evaluate efficacy of anti-covid treatments. Patients who received anti-covid treatments had worse outcomes, as the patients receiving these had more serious COVID-19 disease. Therefore, it was felt not appropriate to present this subgroup analysis.
  4. We agree that this would be useful data to look at, however vaccination data and previous COVID-19 infection is not reliably captured retrospectively from our electronic health records system, therefore due to poor quality and missingness of data, it would be challenging to draw any conclusions on this. We have discussed reasons for potentially better outcomes in our discussion. We also know that in wave 1 there was no COVID-19 vaccine or prior infections in the wave 1 cohort.
  5. Based on your input, we performed an analysis of time from diagnosis to COVID-19 infection.

When evaluating time from cancer diagnosis to COVID-19 diagnosis (in years), we find positive association between longer time to diagnosis and poorer outcomes - HR 1.04 (95% CI 1.00 - 1.07, p-value = 0.03). Incorporating this into our multivariate model, however, its effects become statistically insignificant: 

Variable

HR (95% CI)

p-value

SACT

2.12 (1.12 - 3.98)

0.02

Male

1.5 (0.81 - 2.78)

0.2

Hypertension

1.64 (0.9 - 3)

0.11

Cerebrovascular disease

2.12 (1.01 - 4.46)

0.05

Cardiovascular disease

1.25 (0.67 - 2.36)

0.48

Age (10 years)

1.41 (1.09 - 1.82)

0.01

Wave

0.26 (0.14 - 0.48)

<0.001

Time since cancer diagnosis to COVID-19

1.04 (1 - 1.08)

0.072

The effect of time since cancer diagnosis to COVID-19 as a surrogate measure is likely influenced by collinear variables such as age, SACT and presence of metastatic disease.

  1. All cause mortality data was collection as per methods: “Wave 1 included patients presenting between 1 March 2020 and 31 May 2020 and follow-up data was collected until 12 June 2020. Wave 2 included patients presenting be-tween 1 December 2020 and 8 February 2021 and follow-up data was collected until 18 March 2021”
  2. Additional relevant data is specified in next line as “This included blood results at time of diagnosis of SARS-CoV-2 infection, associated symptoms, and cancer-specific data.”
  3. 60 days after anti-cancer treatment is consistent with our first paper and several large scale COVID-19 and cancer studies also have used the 60 day cut off.

https://www.frontiersin.org/articles/10.3389/fonc.2020.595804/full

https://www.nature.com/articles/s41598-021-84137-5

https://journals.plos.org/plosone/article?id=10.1371/journal.pone.0241261

Data from the CCC-19 consortium (https://www.annalsofoncology.org/article/S0923-7534(21)00874-7) also suggests that the risk is present for patients with cancer whether at less than 1 month or between 1 to 3 months after receipt of chemotherapy.

  1. Instructions in the template are now delated.
  2. It is difficult to present and draw conclusions from symptoms at presentation as due to retrospective nature we noticed that this data was not captured comprehensively enough. However we know from our analysis of wave 1 and wave 2 that severity and symptomology was less in wave 2 as evidenced by percentage of hospitalized patients. Therefore we are using hospitalization as a marker of more severe and symptomatic disease.
  3. Empty columns from template deleted.

Round 2

Reviewer 2 Report

The authors well-addressed all my previous comments. The manuscript can be now accepted for publication after the editorial check.

Reviewer 4 Report

Manuscript was much improved following revisions.